# Antimicrobial Photodynamic Therapy in the Control of COVID-19

**DOI:** 10.3390/antibiotics9060320

**Published:** 2020-06-11

**Authors:** Adelaide Almeida, M. Amparo F. Faustino, Maria G. P. M. S. Neves

**Affiliations:** 1Department of Biology CESAM, University of Aveiro, 3810-193 Aveiro, Portugal; 2Department of Chemistry and LAQV-REQUIMTE, University of Aveiro, 3810-193 Aveiro, Portugal; faustino@ua.pt (M.A.F.F.); gneves@ua.pt (M.G.P.M.S.N.)

**Keywords:** antimicrobial photodynamic therapy, coronaviruses, SARS-CoV-2, COVID-19, control, disinfection

## Abstract

Antimicrobial photodynamic therapy (aPDT), using well known, safe and cost-effective photosensitizers, such as phenothiazines, e.g., methylene blue (MB), or porphyrins, e.g., protoporphyrin-IX (PP-IX), might help to mitigate the COVID-19 either to prevent infections or to develop photoactive fabrics (e.g., masks, suits, gloves) to disinfect surfaces, air and wastewater, under artificial light and/or natural sunlight.

## Main text

The world is facing the Coronavirus disease 2019 (COVID-19) pandemic, it is a highly contagious respiratory illness, without any effective treatment. The spreading of severe acute respiratory syndrome coronavirus 2 (SARS-CoV-2) occurs person-to-person, but also from air droplets, infected objects, and surfaces [1,2,3,4,5]. Consequently, old treatments, such as chloroquine and analogues (e.g., hydroxychloroquine) [6,7,8,9,10] or remdesivir [11,12,13], have been used to treat COVID-19 patients, and old disinfectants (e.g., alcohol-based solutions, hypochlorite solutions) have been applied to surfaces to inactivate the SARS-CoV-2 virus or to disinfect air. The World Health Organization (WHO), the European Centre for Disease Prevention and Control (ECDC) and the United States Centers for Disease Control and Prevention (CDC), among other authorities and entities, have spoken with one voice, emphasizing the utmost importance of hand hygiene, respiratory etiquette, environmental cleaning and disinfection, as well as the importance of maintaining social isolation and increasing the physical distance between people, in order to mitigate SARS-CoV-2 propagation. However, considering the lethality caused by COVID-19 disease, while no vaccine (considered the most effective measure to fight this type of infection) and/or effective new antiviral drugs are available, other approaches should be considered. Moreover, the importance of these measures increases if we think that SARS-CoV-2 is an RNA virus and, as such, has high mutation potential. This means that even if a vaccine is developed in the near future, it may not be effective in the medium/long term.

The use of photodynamic therapy (PDT) can be an alternative approach against SARS-CoV-2 that deserves to be explored. PDT requires the use of a photosensitizer (PS), a molecule that, after being excited by visible light, can react with dioxygen (^3^O_2_, the atmospheric oxygen), producing reactive oxygen species (ROS) such as singlet oxygen (^1^O_2_) and/or superoxide anion, hydroxyl radicals and hydrogen peroxide. These ROS can react with biological molecules (e.g. proteins, lipids and nucleic acids), causing their oxidation and, consequently, damage to cells and tissues. PDT is already approved and routinely used for cancer treatment (e.g., basal cell carcinomas (BCC), esophageal, and lung carcinomas), but also in non-oncological situations like age-related macular degeneration (AMD). Additionally, in the last few years, the approach has been shown to be highly effective against all types of microorganisms, such as Gram-positive and Gram-negative bacteria, fungi, parasites, and viruses [14,15,16,17,18,19,20,21,22,23,24,25,26,27,28,29,30,31,32], a group to which the causative agent of COVID-19 belongs. This broad-spectrum activity will clearly be useful in the treatment of emerging infectious diseases, such as COVID-19 [21,33].

The main advantages of the photodynamic treatment to inactivate microorganisms (designated as antimicrobial photodynamic therapy, (aPDT)) are the lack of microbial specificity and the development of resistance mechanisms. These features are due to the mode of action of aPDT and to the type of biochemical targets affected [34,35,36,37,38,39,40,41]. Although some microorganisms can produce some antioxidant enzymes, such as superoxide dismutase, catalase, and peroxidase, which confer protection against some ROS, this is not true for singlet oxygen (^1^O_2_) [42], which is the main ROS produced by the PSs [14]. Moreover, singlet oxygen has even demonstrated the ability to degrade these antioxidant enzymes [43]. 

The main targets of photodynamic action are the external microbial structures, such as the cell wall, cell membrane, or virus capsid and envelope [14]. As the main targets of aPDT are external structures, the PS does not need to enter the microorganism; the specific and adequate adhesion of the PS to the external structures is sufficient for its destruction when activated by light. In this way, the target microorganisms do not have the opportunity to develop resistance by stopping uptake nor by increasing the metabolic detoxification or the efflux of the drug.

The multi-target nature of aPDT also has a crucial role in minimizing the risk of resistance development, which provides an advantage over conventional antimicrobials. With aPDT, the number of molecular changes needed to ensure survival is very high, requiring that microorganisms develop mutations at various locations to become resistant. These are events less likely to occur than when there is a single site mutation, as in the case of common antimicrobials, namely antivirals and antibiotics. The ROS formed during the photoinactivation process act upon various critical molecular targets, such as proteins, lipids, and nucleic acids [34,44]. Oxidative stress induced by ROS causes irreversible damage to proteins and lipids of the external structures [36,38,45]. However, nucleic acids are only affected by ROS when the microorganisms are already inactivated or non-viable [37,46,47,48]. Thus the possibility that microorganisms can develop resistance mechanisms to this kind of treatment is very low or even absent.

In addition to being active against a wide range of microorganisms, aPDT is effective against drug-resistant microorganisms. In fact, all studies of inactivation of antimicrobial-resistant microorganisms by aPDT have found them to be equally susceptible as their native counterparts [49,50,51,52,53,54,55,56]. Moreover, aPDT has also proven to be effective to degrade the complex matrix of microbial biofilms, the default mode-of-life for many microbial species that cause chronic infections [18]. In fact, aPDT had been recently proposed to combat clinically relevant biofilms such as dental biofilms, ventilator-associated pneumonia, chronic wound infections, oral candidiasis, and chronic rhinosinusitis [57]. As aPDT causes irreversible damage to proteins and lipids, it also affects the expression of virulence factors, such as proteases, alpha-hemolysin, sphingomyelinase, and lipopolysaccharides [58,59,60,61]. The destruction of virulence factors is of extreme importance as they may be secreted by the microorganism during the infection process, but also in the absence of infection, such as in the case of toxins that can cause severe damage to the host.

The rapid PS uptake by the target external structures of the microorganisms when compared with the PS uptake by the host (few minutes versus several hours), and the application of locally targeted illumination, provides a selective therapeutic advantage of this approach when compared to conventional antimicrobials, such as antivirals and antibiotics [14,44].

The photodynamic efficacy of different classes of PSs, namely natural and synthetic tetrapyrrolic macrocycles (e.g., protoporphyrin-IX, chlorophylls, bacteriochlorophylls, *meso*-tetraarylporphyrins, corroles, phthalocyanines), fullerenes, heterocycles like phenothiazinium dyes (e.g., toluidine blue O, methylene blue, and new methylene blue) and psoralens have been evaluated, confirming their role as antimicrobial agents [62]. In fact, some PSs such as the phenothiazine dye methylene blue (MB), Photofrin®, a purified hematoporphyrin mixture, protoporphyrin-IX (PP-IX) and its precursor, 5-aminolevulinic acid (ALA) are molecules already approved for use in cancer treatment, having a well-known safety profile. Moreover, ALA [46] and hematoporphyrin (a derivative of PP-IX) [63,64] are also approved under the context of aPDT, and consequently, these molecules should be promising PSs to inactivate the coronaviruses.

In fact, chloroquine [65,66,67,68], currently used to treat COVID-19, is structurally related to methylene blue. Preliminary data of a recent study suggest that methylene blue might be a good treatment for influenza-like illnesses, such as COVID-19 [69]. In this study, the authors report a cohort of 2500 French patients treated for cancer (breast (40%), lung (20%), prostate (10%), uterine (10%), colon (8%), liver (6%), and miscellaneous (6%) cancers) with a combination of standard therapy and α-lipoic acid, hydroxycitrate and methylene blue (75 mg three times a day). At the date of the study (27 March, 2020), there were no cases of registered COVID-19 or flu-like syndromes in these patients. The use of methylene blue to treat the oncologic patients may have prevented the infection caused by the influenza virus. As stated by the authors [69], methylene blue, besides being considered the ancestor of modern and less toxic anti-malaria drugs, such as chloroquine, may also have other potential therapeutic applications. For instance, it can be used as a neuroprotective agent to facilitate psychotherapeutic interventions in psychiatry and clinical psychology. Moreover, low-level near-infrared light is considered to improve neurological outcomes in humans after ischemic stroke, and also emotional and neurocognitive functions such as sustained attention and working memory [69]. There are also several studies showing that methylene blue, when activated with visible light effectively, inactivates viruses, bacteria, and fungi [70,71,72,73,74,75]. 

As SARS-CoV-2 affects mainly the lower respiratory system, namely the lungs, it is relatively easy to irradiate these internal organs endoscopically using an optical fiber. This can be introduced through the nose, suggesting that aPDT can mediate the inactivation of the virus in the lungs. Indeed, PDT has been approved to treat lung cancer for years and with excellent results [76,77,78,79]. Antimicrobial photodynamic therapy based-nanomaterials (with immobilized PS) have been investigated as novel ways of inhibiting viral, bacterial and fungal infections [62], allowing access to areas with complex anatomy, namely if the nanomaterials are bonded to monoclonal antibodies which could target lung tissue specifically, avoiding less damage to adjacent tissues [78]. Moreover, the coronaviruses group (CoVs) is a quite common virus that is generally associated with upper and lower respiratory tract disorders in humans. Thus, aPDT can prevent the entry of respiratory viruses, such as the SARS-CoV-2, other coronavirus strains, and also other respiratory viruses through the upper respiratory tract system if nostrils are irradiated after the addition of a PS. In fact, there are already some applications to inactivate bacteria in the nostrils by aPDT [80].

Relative to the disinfection of objects, including fabrics and surfaces, besides artificial white light, natural sunlight can be used as a light source to inactivate the coronavirus. The use of sunlight as the light source turns out to be an inexpensive aPDT procedure since it is based on the use of a low cost and worldwide-available visible light source. In addition, photosensitizers like MB or PP-IX based derivatives do not accumulate in the environment since they are degraded by exposure to sunlight, although they are able, in the meantime, to inactivate the microorganisms [14,62,81].

Recently, it has been possible to detect the presence of SARS-CoV-2 in wastewater in different countries [82,83,84,85]. This result opened up a range of opportunities in the investigation of the SARS-CoV-2 virus in wastewater in order to monitor and mitigate the spread of COVID-19 in the community. For instance, it is expected that the detection of the SARS-CoV-2 in wastewater can serve as a warning of a new outbreak of the disease, and consequently, several countries are already investigating this issue. As these studies show that the SARS-CoV-2 can be identified in the wastewaters, sewers can represent a source of coronavirus environment contamination. In fact, it is well known that wastewater, with a high content of pathogenic microorganisms, including viruses, is a current area of concern affecting the quality of natural receiving waters, and consequently, human health.

The conventional approach to treat wastewater, including primary, secondary, and tertiary treatment, consists of a combination of physical, chemical, and biological processes. Usually, wastewater from urban areas is secondarily, rarely tertiary, treated. Secondary treatment of wastewater is usually considered sufficient, although the secondary effluent still contains infective concentrations of microorganisms. Moreover, the emerging pathogens, such as the SARS-CoV-2, brought serious risks when wastewater is not properly treated [86], contributing to a wide spread of emerging microorganisms [87]. In order to reduce the concentration of pathogens in wastewater, the secondarily treated effluent is usually subjected to disinfection [88] with chlorine (the main used method around the world), ozone (O_3_), or ultraviolet light (UV) [81]. Chlorine (Cl_2_) and ozone may lead to the formation of toxic products, such as monochloramine; UV can cause modifications in microorganism, contributing to the selection of resistant genes; ozone and UV require equipment maintenance and replacement costs [89,90,91]. As the traditional tertiary disinfection treatments used to reduce microorganism concentration may be expensive, toxic to aquatic species, and induce genetic damage in microorganisms, the development of new technologies for wastewater decontamination is also urgent.

Even though it is not yet clear whether the SARS-CoV-2 is viable in wastewater for a long period, which could facilitate fecal–oral transmission, it is of utmost importance to develop wastewater treatment processes that effectively inactivate this highly transmissible virus. According to the literature, other RNA viruses are frequently transmitted to humans through wastewater contamination [92]. Since aPDT is truly antimicrobial, it is expected to be effective against emerging/unknown pathogens, being an alternative approach to inactivate the coronaviruses in wastewater.

The promising results of aPDT on inactivation of viruses, namely against those with an envelope, like the coronaviruses [29,33], accompanied by the possibility to immobilize easily prepared PSs on insoluble inert materials such as nano/microparticles, films and polymers [28,31,93,94,95,96] suggest that this principle can be applied to wastewater treatment to inactivate microorganisms, including SARS-CoV-2. According to the literature, different PSs already proved to be effective to photoinactivate microorganisms, including viruses, in samples of secondary-treated wastewater [97,98,99]. These studies were performed at low PS concentrations (5–10 μM) under low artificial white light irradiance (380–700 nm, 4 mW cm^−2^) or under natural sunlight conditions. Moreover, magnetic nanoparticles with different immobilized porphyrins were reported, and studies showed that these photoactive materials can be easily recovered from the water matrix for subsequent reuse, maintaining their effectiveness against viruses under white light irradiation [28,34].

In addition, the immobilization of PSs in synthetic and natural fibers like cellulose and other analogues can afford textile products with effective antimicrobial features. This is considered of high relevance for the health industry in order to maintain the availability of masks, gowns, bedding, and wound dressings. With this technology, it is expected to minimize the occurrence of hospital-acquired infections and to protect healthcare workers like doctors and nurses [62].

The studies mentioned show that aPDT is an excellent approach to inactivate a large spectrum of pathogenic microorganisms, namely those resistant to conventional antimicrobials under clinical and non-clinical contexts. However, for all the knowledge already gained in aPDT to be translated to more practical applications in the context of the current pandemic situation, it is fundamental and urgent that there be more collaboration among basic researchers (e.g., chemists, biochemists, biologists, materials engineers) and the technologists, clinicians, and pharmaceutical companies corroborated with financial support from national and international institutions. In this way, the excellent research behind aPDT can contribute to combating multidrug-resistant microorganisms and viruses, and be one of the frontline tools to prevent new pandemic crises. 

In conclusion, aPDT treatment using well known, safe, and cost-effective PSs, such as MB or PP-IX, might help to mitigate the COVID-19 either to treat infected patients, to develop functional photoactive textiles, to auto-disinfect surfaces or to disinfect water and air (Figure 1).

## Figures and Tables

**Figure 1 antibiotics-09-00320-f001:**
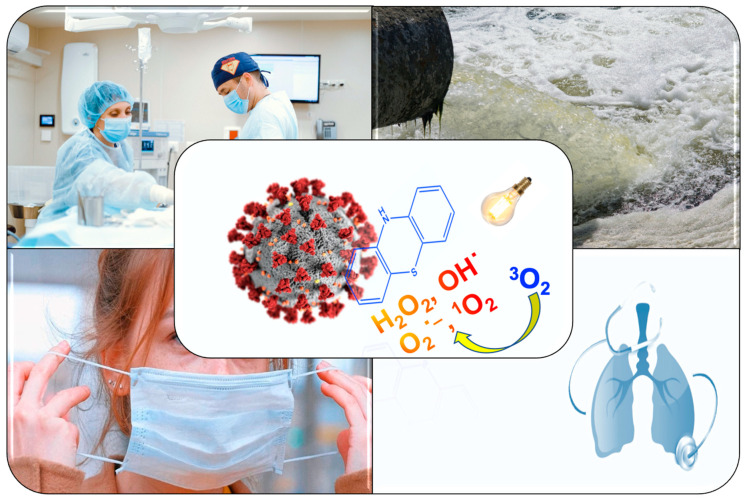
Potential applications of antimicrobial PhotoDynamic Therapy (aPDT) against the severe acute respiratory syndrome coronavirus 2 (SARS-CoV-2) in different contexts.

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
