# Peer review of "Antimicrobial Photodynamic Therapy in the Control of COVID-19"

_antibiotics, 2020, doi:10.3390/antibiotics9060320_

Round 1

Reviewer 1 Report

Minor

  1. Author must spell out the full form of abbreviations in on their first appearance (eg. ROS and others)
  2. Typographical errors here and there.

Author Response

Dear Reviewer,

Thank you for your amendments to the manuscript “Antimicrobial photodynamic therapy in the control of COVID-19” – Manuscript ID antibiotics-8057468. I am pleased to inform you that all your comments and suggestions were analysed in this letter as well as in our manuscript (highlighted in yellow).

Reviewer comments:

 Reviewer #1

  1. Author must spell out the full form of abbreviations in on their first appearance (eg. ROS and others).

We thank the reviewer remark. The abbreviations “ROS” and “PS” were spell out in the full form in on their first appearance.

  1. Typographical errors here and there.

The typos were corrected throughout the manuscript.

Reviewer 2 Report

The authors should incorporate a figure, representing various applicability of the PDTs along with a mechanistic/strategic diagram for the treatment of SARS-CoV-2.

Author Response

Dear Reviewer,

Thank you for your amendment to the manuscript “Antimicrobial photodynamic therapy in the control of COVID-19” – Manuscript ID antibiotics-8057468. I am pleased to inform you that your comment was analysed in this letter as well as in our manuscript (highlighted in yellow).

Reviewer comment:

Reviewer #2

  1. The authors should incorporate a figure, representing various applicability of the PDTs along with a mechanistic/strategic diagram for the treatment of SARS-CoV-2.

We thank the reviewer for the suggestion and a figure representing the potential applicability of aPDT against the SARS-CoV-2 under different contexts was added to the manuscript.

Reviewer 3 Report

The manuscript n° antibiotics-805746 proposed by A. Almeida and coll. for publication as a review (or “current opinion”) in Antibiotics, is entitled “Antimicrobial photodynamic therapy in the control 2 of COVID-19 ». In this article, now well described Antimicrobial PhotoDynamic Therapy (aPDT), using well known photosensitizers (PSs) is envisaged in order to « mitigate the COVID-19 either to prevent infections or to develop photoactive fabrics to disinfect surfaces, air and wastewater » under light. The studies mentioned in the manuscript show that aPDT can be envisaged as an excellent approach to inactivate a large spectrum of pathogenic microorganisms namely those resistant to conventional antimicrobials under clinical and non-clinical contexts; of course, potential anti-virus application was already communicated. The short review is quite clear and well written, but actually, the authors are not the only one in the domain thinking to such an application and opportunity for the aPDT; there are also not the first one to publish about this idea (see Photobiomodulation, Photomedicine, and Laser Surgery, 2020, 38, 255–257, DOI: 10.1089/photob.2020.4868 for example).

In such an SHORT review (or opinion), it is of course difficult to be exhaustive, nevertheless, I think several bibliographic references are missing :

Recent reviews on aPDT, for example Cieplik et al., 2018; Hu et al., 2018; Tomb et al., 2018; Wozniak and Grinholc, 2018 ; in this domain of aPDT, G. Gasser has also to be cited. The same remark concerning T. Le Gall et al, 2018 and the use of Ru(II) complexes for a A “Structure-Activity Study on Clinical Bacterial Strains”; insert for example lines 70-72.

Studies from Pr. D. Raoult about hydroxychloroquine have to be cited (ref 6-9).

Please take care to VERY short paragraphs, and several too long sentences !!

To conclude, this short review, or opinion paper is interesting and quite well written. I need the authors to answer the remarks done before a re-evaluation for a potential publication in Antibiotics.

Author Response

Dear Reviewer,

Thank you for your amendments addressed to the manuscript “Antimicrobial photodynamic therapy in the control of COVID-19” – Manuscript ID antibiotics-8057468. I am pleased to inform you that all the comments and suggestions of the referees were analysed in this letter as well as in our manuscript (highlighted in yellow).

Reviewer comments:

Reviewer #3

  1. The manuscript n° antibiotics-805746proposed by A. Almeida and coll. for publication as a review (or “current opinion”) in Antibiotics, is entitled “Antimicrobial photodynamic therapy in the control 2 of COVID-19 ». In this article, now well described Antimicrobial PhotoDynamic Therapy (aPDT), using well known photosensitizers (PSs) is envisaged in order to « mitigate the COVID-19 either to prevent infections or to develop photoactive fabrics to disinfect surfaces, air and wastewater » under light. The studies mentioned in the manuscript show that aPDT can be envisaged as an excellent approach to inactivate a large spectrum of pathogenic microorganisms namely those resistant to conventional antimicrobials under clinical and non-clinical contexts; of course, potential anti-virus application was already communicated. The short review is quite clear and well written, but actually, the authors are not the only one in the domain thinking to such an application and opportunity for the aPDT; there are also not the first one to publish about this idea (see Photobiomodulation, Photomedicine, and Laser Surgery202038, 255–257, DOI: 10.1089/photob.2020.4868 for example).

In order to update the state of the art, the meanwhile published paper suggested by the reviewer was added to the manuscript.

  1. In such an SHORT review (or opinion), it is of course difficult to be exhaustive, nevertheless, I think several bibliographic references are missing : Recent reviews on aPDT, for example Cieplik et al., 2018; Hu et al., 2018; Tomb et al., 2018; Wozniak and Grinholc, 2018; in this domain of aPDT, G. Gasser has also to be cited. The same remark concerning T. Le Gall et al, 2018 and the use of Ru(II) complexes for a A “Structure-Activity Study on Clinical Bacterial Strains”; insert for example lines 70-72. Studies from Pr. D. Raoult about hydroxychloroquine have to be cited (ref 6-9).

Following the reviewer suggestions the recent reviews as well as the recent research papers suggested were added to the manuscript.

  1. Please take care to VERY short paragraphs, and several too long sentences !!

According with the suggestion, the very short paragraphs and too long sentences were rewritten.

  1. To conclude, this short review, or opinion paper is interesting and quite well written. I need the authors to answer the remarks done before a re-evaluation for a potential publication in Antibiotics.

We appreciate the reviewer observation.

Round 2

Reviewer 3 Report

The authors made the expected modifications to the manuscript; I think it now deserves for publication in Antibiotics

Author Response

The proposed changes to English improved the manuscript quality and consequently were accepted by the authors. Anew version was uploaded.

Kind regards

Adelaide Almeida